# LORENTZIAN DISTANCE LEARNING

## ABSTRACT

This paper introduces an approach to learn representations based on the Lorentzian distance in hyperbolic geometry. Hyperbolic geometry is especially suited to hierarchically-structured datasets, which are prevalent in the real world. Current hyperbolic representation learning methods compare examples with the Poincaré distance metric. They formulate the problem as minimizing the distance of each node in a hierarchy with its descendants while maximizing its distance with other nodes. This formulation produces node representations close to the centroid of their descendants. We exploit the fact that the centroid w.r.t the squared Lorentzian distance can be written in closed-form. We show that the Euclidean norm of such a centroid decreases as the curvature of the hyperbolic space decreases. This property makes it appropriate to represent hierarchies where parent nodes minimize the distances to their descendants and have smaller Euclidean norm than their children. Our approach obtains state-of-the-art results in retrieval and classification tasks on different datasets.

## 1 INTRODUCTION

Generalizations of Euclidean space are important forms of data representation in machine learning. For instance, kernel methods rely on Hilbert spaces that possess the structure of the inner product and can therefore be used to compare examples. The properties of such spaces are well-known and closed-form relations are often exploited to obtain efficient, scalable, and interpretable training algorithms. While representing examples in a Euclidean space is appropriate to compare lengths and angles, non-Euclidean representations are useful when the task requires specific structure.

A common and natural non-Euclidean representation space is the spherical model (*e.g.* (Wang et al., 2017)) where the data lies on a unit hypersphere $\mathcal{S}^d = \{\mathbf{x} \in \mathbb{R}^d : \|\mathbf{x}\| = 1\}$ and angles are compared with the cosine similarity function. Recently, some machine learning approaches (Nickel & Kiela, 2017; 2018; Ganea et al., 2018) have considered representing hierarchical datasets with the hyperbolic model. The motivation is that any finite tree can be mapped into a finite hyperbolic space while approximately preserving distances (Gromov, 1987). Since hierarchies can be formulated as trees, hyperbolic spaces can be used to represent hierarchically structured data where the high-level nodes of the hierarchy are represented close to the origin (*i.e.* with small Euclidean norm) whereas leaves are further away from the origin.

Since their first formulation in the early nineteenth century by Lobachevsky and Bolyai, hyperbolic spaces have been used in many domains. In particular, they became popular in mathematics (*e.g.* space theory and differential geometry (Ratcliffe, 2006)), and physics when Varicak (1908) discovered that special relativity theory (Einstein, 1905) had a natural interpretation in hyperbolic geometry. Various hyperbolic geometries and related distances have been studied since then. Among them are the Poincaré metric, the Lorentzian distance (Ratcliffe, 2006), and the gyrodistance (Ungar, 2010).

In the case of hierarchical datasets, machine learning approaches that learn hyperbolic representations designed to preserve the hierarchical similarity order have typically employed the Poincaré metric. Usually, the optimization problem is formulated so that the representation of a node in a hierarchy should be closer to the representation of its children and other descendants than to any other node in the hierarchy. Based on Gromov (1987), the Poincaré metric is a sensible dissimilarity function as it satisfies all the properties of a distance metric and is thus natural to interpret.

In this paper, we propose to exploit the squared Lorentzian distance instead of the usual Poincaré distance. The squared Lorentzian distance is linear in the hyperbolic representations and differentiable

on the whole domain, which avoids numerical instabilities and exploding gradients. Moreover, the order of the Poincaré distances between pairs of points is the same as the order of the (squared) Lorentzian distances in the unit hyperboloid model. We may therefore directly adapt existing approaches by replacing Poincaré distances with squared Lorentzian distances. The main property of interest is that the center of mass *w.r.t.* the squared Lorentzian distance can be written in closed form and is then easy to interpret. In particular, we show that the Euclidean norm of the center of mass of a set of different points decreases as the curvature of the exploited hyperboloid model[1] decreases and tends to $-\infty$. This property makes it ideal to represent hierarchical structures since we would like the root of a subtree to minimize the distance to all of its descendants (*i.e.* it should be their centroid *w.r.t.* to the chosen distance function) and have smaller Euclidean norm than them.

Our contributions are both analytic and experimental. We study properties of the chosen dissimilarity function and explain why they make it appropriate for the task of interest. In particular, we show that the Euclidean norm of the center of mass in the Poincaré ball representation decreases as the curvature $-1/\beta \in \mathbb{R}_-$ of the hyperboloid model decreases (*i.e.* as $\beta > 0$ decreases). Curvature can then be used as a hyperparameter to implicitly enforce the root of a subtree to have smaller Euclidean norm than the other nodes. We experimentally show on different problems that exploiting the squared Lorentzian distance improves retrieval and classification performance.

## 2 BACKGROUND

In this section, we provide some technical background about hyperbolic geometry and introduce relevant notation. The interested reader may refer to (Ratcliffe, 2006) for more detail.

### 2.1 NOTATION AND DEFINITIONS

To simplify the notation, we consider that vectors are row vectors and $\|\cdot\|$ is the $\ell_2$-norm. In the following, we consider three important spaces.

**Poincaré ball:** The Poincaré ball $\mathcal{P}^d$ is defined as the set of $d$-dimensional vectors with Euclidean norm smaller than 1 (*i.e.* $\mathcal{P}^d = \{\mathbf{x} \in \mathbb{R}^d : \|\mathbf{x}\| < 1\}$). Its associated distance is the Poincaré distance metric defined in Eq. (3).

**Hyperboloid model:** We consider some specific hyperboloid models $\mathcal{H}^{d,\beta}$ defined as follows:

$$\mathcal{H}^{d,\beta} := \{\mathbf{a} = (a_0, \cdots, a_d) \in \mathbb{R}^{d+1} : \|\mathbf{a}\|_{\mathcal{L}}^2 = -\beta, a_0 > 0, \beta > 0\} \tag{1}$$

where $\|\mathbf{a}\|_{\mathcal{L}}^2 = \langle \mathbf{a}, \mathbf{a} \rangle_{\mathcal{L}}$ is the *squared Lorentzian norm* of $\mathbf{a}$. The squared Lorentzian norm is derived from the *Lorentzian inner product* defined as:

$$\forall \mathbf{a} = (a_0, \cdots, a_d) \in \mathcal{H}^{d,\beta}, \mathbf{b} = (b_0, \cdots, b_d) \in \mathcal{H}^{d,\beta}, \langle \mathbf{a}, \mathbf{b} \rangle_{\mathcal{L}} := -a_0 b_0 + \sum_{i=1}^{d} a_i b_i \leq -\beta \tag{2}$$

It is worth noting that $\langle \mathbf{a}, \mathbf{b} \rangle_{\mathcal{L}} = -\beta$ iff $\mathbf{a} = \mathbf{b}$. Otherwise, $\langle \mathbf{a}, \mathbf{b} \rangle_{\mathcal{L}} < -\beta$ for all pair $(\mathbf{a}, \mathbf{b}) \in (\mathcal{H}^{d,\beta})^2$. Vectors in $\mathcal{H}^{d,\beta}$ are a subset of positive time-like vectors[2]. Moreover, every vector $\mathbf{a} \in \mathcal{H}^{d,\beta}$ satisfies $a_0 = \sqrt{\beta + \sum_{i=1}^{d} a_i^2}$. We note $\mathcal{H}^d := \mathcal{H}^{d,1}$ the space obtained when $\beta = 1$; it is usually called the (positive sheet of the) *unit hyperboloid* model and is the main hyperboloid model considered in the literature.

**Model space:** Finally, we note $\mathcal{F}^d \subseteq \mathbb{R}^d$ the output space of our model (*e.g.* the output representation of some linear model or neural network). In the following, we consider that $\mathcal{F}^d = \mathbb{R}^d$.

### 2.2 OPTIMIZING THE POINCARÉ DISTANCE METRIC

Most methods that compare hyperbolic representations (Nickel & Kiela, 2017; 2018; Ganea et al., 2018; Gulcehre et al., 2018) consider the Poincaré distance metric defined as:

$$\forall \mathbf{c} \in \mathcal{P}^d, \mathbf{d} \in \mathcal{P}^d, \mathrm{d}_{\mathcal{P}}(\mathbf{c}, \mathbf{d}) = \mathrm{arcosh}\left(1 + 2\frac{\|\mathbf{c} - \mathbf{d}\|^2}{(1 - \|\mathbf{c}\|^2)(1 - \|\mathbf{d}\|^2)}\right) \tag{3}$$

---

[1] The curvature of the considered hyperboloid models is negative and is -1 for the unit hyperboloid model.

[2] A vector $\mathbf{a}$ that satisfies $\langle \mathbf{a}, \mathbf{a} \rangle_{\mathcal{L}} < 0$ is called time-like and it is called positive iff $a_0 > 0$.

which satisfies all the properties of a distance metric and is therefore natural to interpret. Direct optimization of problems using the distance formulation in Eq. (3) is numerically instable for two main reasons (see for instance (Nickel & Kiela, 2018) or (Ganea et al., 2018, Section 4)). First, the denominator depends on the norm of examples, so optimizing over $\mathbf{c}$ and $\mathbf{d}$ when either of their norms is close to 1 leads to numerical instability. Second, elements have to be re-projected onto the Poincaré ball at each iteration with a fixed maximum norm.

To increase the numerical stability of their solver, Nickel & Kiela (2018) propose to use an equivalent formulation of $\mathrm{d}_{\mathcal{P}}$ in the unit hyperboloid model. They use the fact that there exists an invertible mapping $h : \mathcal{H}^{d,\beta} \to \mathcal{P}^d$ defined as:

$$\forall \mathbf{a} = (a_0, \cdots, a_d) \in \mathcal{H}^{d,\beta}, \ h(\mathbf{a}) := \frac{1}{1 + \sqrt{1 + \sum_{i=1}^{d} a_i^2}}(a_1, \cdots, a_d) \in \mathcal{P}^d \tag{4}$$

When $\beta = 1$, we have the following equivalence:

$$\forall \mathbf{a} \in \mathcal{H}^d, \mathbf{b} \in \mathcal{H}^d, \ \mathrm{d}_{\mathcal{H}}(\mathbf{a}, \mathbf{b}) = \mathrm{d}_{\mathcal{P}}(h(\mathbf{a}), h(\mathbf{b})) = \mathrm{arcosh}\left(-\langle \mathbf{a}, \mathbf{b}\rangle_{\mathcal{L}}\right) \tag{5}$$

which corresponds to the (Lorentzian) angle between $\mathbf{a}$ and $\mathbf{b}$. It is shown in (Nickel & Kiela, 2018) that optimizing the formulation in Eq. (5) is more stable numerically. Nonetheless, the gradient of Eq. (5) still tends to $+\infty$ as $\mathrm{d}_{\mathcal{H}}(\mathbf{a}, \mathbf{b})$ tends to 0. Interestingly, one can observe from Eq. (5) that preserving the order of Poincaré distances is equivalent to preserving the order of the opposite of Lorentzian inner products since the *arcosh* function is monotonically increasing on its domain $[1, +\infty)$. In the same way as kernel methods that consider inner products in Hilbert spaces as similarity measures, we consider in this paper the Lorentzian inner product.

## 3 PROPOSED METHOD

We present in this section the (squared) *Lorentzian distance function* $\mathrm{d}_{\mathcal{L}}$ (Ratcliffe, 2006) and mention the properties that make it appropriate for our task. We give the formulation of its center of mass and observe that its Euclidean norm depends on the curvature $-1/\beta$ which can then be used as a hyper-parameter of the model to implicitly enforce centroids to have small Euclidean norm. We also discuss optimization details.

### 3.1 LORENTZIAN DISTANCE AND MAPPINGS

The squared Lorentzian distance (Ratcliffe, 2006) is defined as:

$$\forall \mathbf{a} \in \mathcal{H}^{d,\beta}, \mathbf{b} \in \mathcal{H}^{d,\beta}, \ \mathrm{d}_{\mathcal{L}}^2(\mathbf{a}, \mathbf{b}) = \|\mathbf{a} - \mathbf{b}\|_{\mathcal{L}}^2 = \|\mathbf{a}\|_{\mathcal{L}}^2 + \|\mathbf{b}\|_{\mathcal{L}}^2 - 2\langle \mathbf{a}, \mathbf{b}\rangle_{\mathcal{L}} = -2\beta - 2\langle \mathbf{a}, \mathbf{b}\rangle_{\mathcal{L}} \tag{6}$$

It satisfies all the axioms of a distance metric except the triangle inequality (*i.e.* non-negativity, symmetry, identity of indiscernibles). One interesting property is that $\mathrm{d}_{\mathcal{L}}^2$ is linear in both $\mathbf{a}$ and $\mathbf{b}$. Moreover, when $\beta = 1$, it defines the hyperbolic length of a curve in $\mathcal{H}^d$ (Ratcliffe, 2006).

**Mapping:** Current hyperbolic machine learning models re-project at each iteration their learned representations onto the Poincaré ball (Nickel & Kiela, 2017) or use some exponential map (Nickel & Kiela, 2018; Gulcehre et al., 2018) to directly optimize on the unit hyperboloid model (*i.e.* when $\beta = 1$). Since our approach does not necessarily consider that $\beta = 1$, we consider the invertible mapping $g_\beta : \mathcal{F}^d \to \mathcal{H}^{d,\beta}$ defined as:

$$\forall \mathbf{f} = (f_1, \cdots, f_d) \in \mathcal{F}^d, \ g_\beta(\mathbf{f}) := (\sqrt{\|\mathbf{f}\|^2 + \beta}, f_1, \cdots, f_d) \in \mathcal{H}^{d,\beta} \tag{7}$$

As mentioned in Section 2.1, $\mathcal{F}^d$ is the space of our model; it is $\mathbb{R}^d$ in practice. To be fair with baselines, the examples in $\mathcal{F}^d$ in our experiments are non-parametric embeddings. $\mathcal{F}^d$ may also be the output space of some parametric model such as a neural network.

We compare two examples $\mathbf{f}_1 \in \mathcal{F}^d$ and $\mathbf{f}_2 \in \mathcal{F}^d$ with $\mathrm{d}_{\mathcal{L}}^2$ by calculating:

$$\forall \mathbf{f}_1 \in \mathcal{F}^d, \mathbf{f}_2 \in \mathcal{F}^d, \ \mathrm{d}_{\mathcal{L}}^2(g_\beta(\mathbf{f}_1), g_\beta(\mathbf{f}_2)) = -2\beta - 2\langle g_\beta(\mathbf{f}_1), g_\beta(\mathbf{f}_2)\rangle_{\mathcal{L}} \tag{8}$$

$$= -2\left[\beta + \langle \mathbf{f}_1, \mathbf{f}_2\rangle - \sqrt{\|\mathbf{f}_1\|^2 + \beta}\sqrt{\|\mathbf{f}_2\|^2 + \beta}\right] \tag{9}$$

**Preserved order of Euclidean norms:** Although examples are compared with $d_{\mathcal{L}}^2$ in the hyperbolic space $\mathcal{H}^{d,\beta}$ where all the points have the same Lorentzian norm, it is worth noting that the order of the Euclidean norms of examples is preserved along the three spaces $\mathcal{F}^d$, $\mathcal{H}^{d,\beta}$ and $\mathcal{P}^d$ with the mappings $g_\beta$ and $h$. The preservation with $g_\beta$ is straightforward: $\forall \mathbf{f}_1, \mathbf{f}_2 \in \mathcal{F}^d, \|\mathbf{f}_1\| < \|\mathbf{f}_2\| \iff \sqrt{2\|\mathbf{f}_1\|^2 + \beta} = \|g_\beta(\mathbf{f}_1)\| < \|g_\beta(\mathbf{f}_2)\|$. The proof of the following theorem is given in the appendix:

**Theorem 3.1** (Order of Euclidean norms). *The following order of norms is preserved with $h \circ g$:*

$$\forall \mathbf{a} \in \mathcal{F}^d, \mathbf{b} \in \mathcal{F}^d, \|\mathbf{a}\| < \|\mathbf{b}\| \iff \|h(g_\beta(\mathbf{a}))\| < \|h(g_\beta(\mathbf{b}))\| \tag{10}$$

In conclusion, the Euclidean norms of examples can be compared equivalently in any space. This is particularly useful if we want to study the Euclidean norm of centroids *w.r.t.* examples, and also understand the significance of the following section.

## 3.2 CENTROID PROPERTIES

We now study the center of mass *w.r.t.* the squared Lorentzian distance. Ideally, we would like the centroid of node representations to be (close to) the representation of their least common ancestor.

**Lemma 3.2** (Center of mass of the Lorentzian inner product). *the point $\boldsymbol{\mu} \in \mathcal{H}^{d,\beta}$ that maximizes the following problem $\max_{\boldsymbol{\mu} \in \mathcal{H}^{d,\beta}} \sum_{i=1}^n \nu_i \langle \mathbf{x}_i, \boldsymbol{\mu} \rangle_{\mathcal{L}}$ where $\forall i, \mathbf{x}_i \in \mathcal{H}^{d,\beta}, \forall i, \nu_i \geq 0, \sum_i \nu_i > 0$ is:*

$$\boldsymbol{\mu} = \sqrt{\beta} \frac{\sum_{i=1}^n \nu_i \mathbf{x}_i}{|\|\sum_{i=1}^n \nu_i \mathbf{x}_i\|_{\mathcal{L}}|} = \sqrt{\beta} \frac{\sum_{i=1}^n \nu_i \mathbf{x}_i}{\sqrt{-\|\sum_{i=1}^n \nu_i \mathbf{x}_i\|_{\mathcal{L}}^2}} \tag{11}$$

where $|\|\mathbf{a}\|_{\mathcal{L}}|$ is the modulus of the imaginary Lorentzian norm of the positive time-like vector $\mathbf{a}$. The proof is given in the appendix.

**Theorem 3.3** (Centroid of the squared Lorentzian distance). *The point $\boldsymbol{\mu} \in \mathcal{H}^{d,\beta}$ that minimizes the problem $\min_{\boldsymbol{\mu} \in \mathcal{H}^{d,\beta}} \sum_{i=1}^n \nu_i d_{\mathcal{L}}^2(\mathbf{x}_i, \boldsymbol{\mu})$ where $\forall i, \mathbf{x}_i \in \mathcal{H}^{d,\beta}, \nu_i \geq 0, \sum_i \nu_i > 0$ is given in Eq. (11)*

The proof exploits the formulation given in Eq. (6). From Eq. (11), one can see that the centroid of one example is the example itself.

**Theorem 3.4.** *The Euclidean norm in $\mathcal{P}^d$ of the center of mass of a set of different examples decreases as $\beta > 0$ decreases.*

The proof is given in the appendix. Fig. 1 illustrates the 2-dimensional Poincaré ball representation of the centroid of a set of 10 different points *w.r.t.* to the Poincaré metric, the gyrodistance (Ungar, 2010), and the squared Lorentzian distance for different values of $\beta > 0$. The Euclidean norm in $\mathcal{P}^d$ of the centroid does decrease as $\beta$ decreases.

We provide in the following some side remarks that are useful to understand the behavior of the Lorentzian distance. In particular, we explain its behavior in extreme cases when $\beta$ tends to 0.

**Theorem 3.5** (Nearest point of the squared Lorentzian distance). *The point $\mathbf{a}_j \in \mathcal{A} = \{\mathbf{a}_k \in \mathcal{H}^{d,\beta}\}_{k=1}^m$ that minimizes the problem $\min_{\mathbf{a}_j \in \mathcal{A}} \sum_{i=1}^n d_{\mathcal{L}}^2(\mathbf{x}_i, \mathbf{a}_j)$ where $\forall i, \mathbf{x}_i \in \mathcal{H}^{d,\beta}$ is the example in $\mathcal{A}$ that is the closest to the centroid of $\mathbf{x}_1, \cdots, \mathbf{x}_n$ w.r.t. the squared Lorentzian distance.*

The theorem can be used to compute medoids. It shows that distances with a set of points can be compared with only one point which is the centroid. The main interest of this theorem in our case is that it helps understand the significance of studying the properties of the centroid *w.r.t.* the Lorentzian distance. Interpreting distances to a set of points can be done by interpreting their centroid.

Moreover, from the Cauchy-Schwarz inequality, as $\beta$ tends to 0, the Lorentzian distance in Eq. (9) to $\mathbf{f}_1$ tends to 0 for any vector $\mathbf{f}_2$ that can be written $\mathbf{f}_2 = \tau \mathbf{f}_1$ with $\tau \geq 0$. The distance is greater otherwise. Therefore, the Lorentzian distance with a set of different points tends to be smaller along the ray that contains elements that can be written $\tau \boldsymbol{\mu}$ where $\tau \geq 0$ and $\boldsymbol{\mu}$ is their centroid as can be seen in Fig. 3.

**Decreasing the curvature in the Poincaré metric formulation?** As defined in Eq. (5), the curvature of the hyperboloid model cannot be decreased in the Poincaré metric. Indeed, the domain of arcosh is $[1, +\infty)$, and the values of the opposite of the Lorentzian inner product are in the interval $[\beta, +\infty)$, which is why $\beta$ has to be greater than or equal to 1 when using the Poincaré metric.

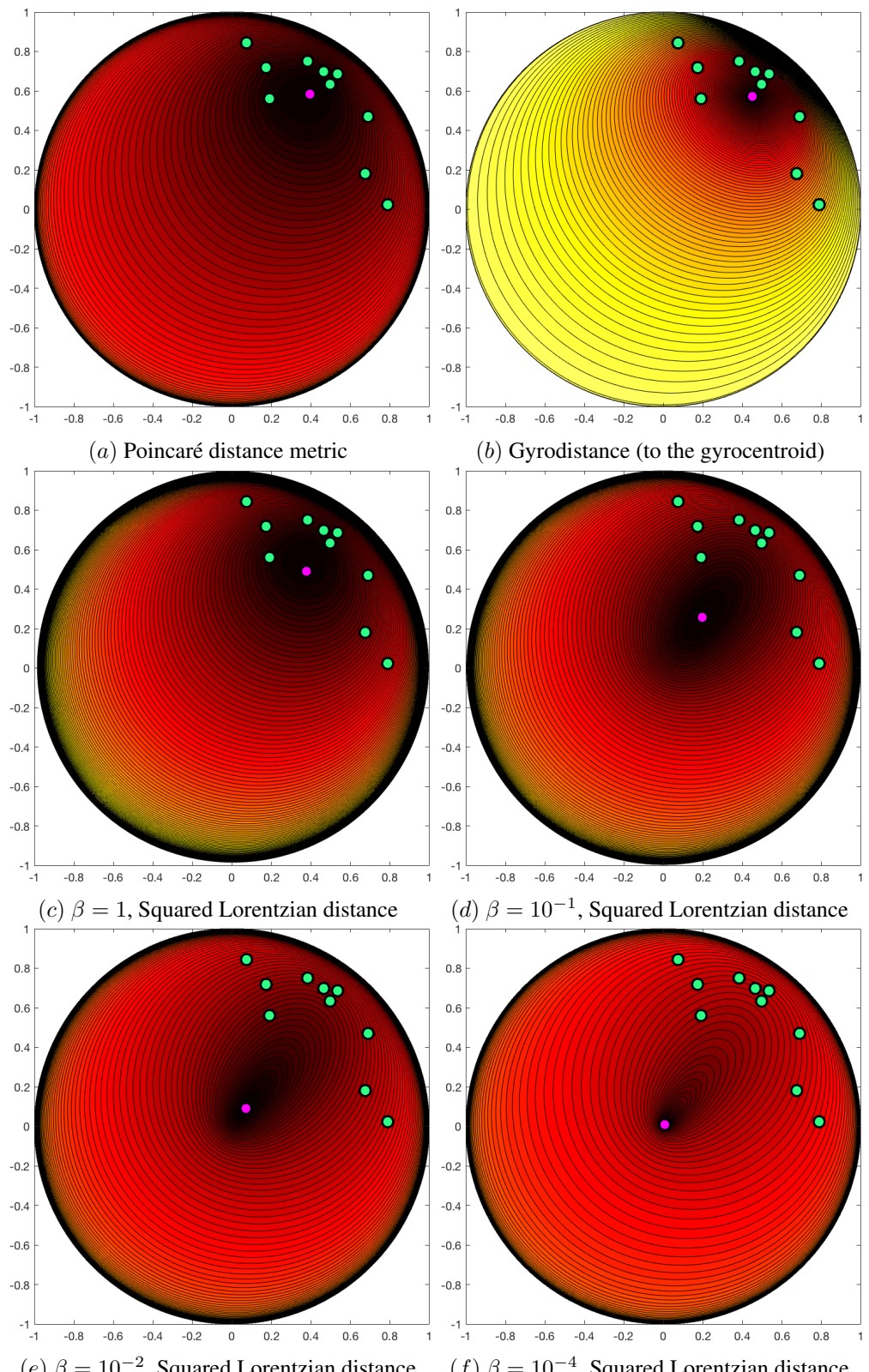

Figure 1: 10 examples are represented in green in a Poincaré ball. Their centroid *w.r.t.* the chosen distance function is in magenta and the level sets represent the sum of the distances between the current point and the training examples. The centroid *w.r.t.* the squared Lorentzian distance can be calculated in closed-form. Smaller values of $\beta$ induce smaller Euclidean norms of the centroid.

### 3.3 Optimization problem and solver

The primary contribution of this paper is the study of the squared Lorentzian distance to represent hierarchically-structured datasets. Even if models that exploit it do not optimize some centroid-based approach such as (Law et al., 2017; Snell et al., 2017), our analysis provides tools to interpret distances between sets of similar points.

We replace the Poincaré metric used by current hyperbolic representation learning models (Nickel & Kiela, 2017; 2018) with the squared Lorentzian distance. Unfortunately, none of these approaches formulate their constraints so that the common ancestor of a set of nodes is their centroid. Nonetheless, they try to minimize the distances between similar examples to preserve orders of distances so that ancestors are closer to their descendants than to unrelated nodes. This indirectly enforces an example similar to a set of examples to be close to their centroids. We briefly describe the constraints used in Nickel & Kiela (2017), though we note that the Lorentzian distance can be exploited in any metric learning problem.

**Constraints:** Nickel & Kiela (2017) consider subsumption relations in a hierarchy (*e.g.* WordNet nouns). They consider the set $\mathcal{D} = \{(u, v)\}$ such that each node $u$ is an ancestor of $v$ in the tree. For each node $u$, a set of negative nodes $\mathcal{N}(u)$ is created: each example in $\mathcal{N}(u)$ is not a descendant of $u$ in the tree. Their problem is then formulated as learning the representations $\{\mathbf{f}_i\}_{i=1}^n$ that minimize:

$$L_{neighborhood} = -\sum_{(u,v)\in\mathcal{D}} \log \frac{e^{-\mathsf{d}(\mathbf{f}_u,\mathbf{f}_v)}}{\sum_{v'\in\mathcal{N}(u)\cup\{v\}} e^{-\mathsf{d}(\mathbf{f}_u,\mathbf{f}_{v'})}} \tag{12}$$

where d is the chosen distance function. The problem is optimized via projected gradient-based method. We replace $\mathsf{d}(\mathbf{f}_u, \mathbf{f}_v)$ by $\mathsf{d}_{\mathcal{L}}^2(g_\beta(\mathbf{f}_u), g_\beta(\mathbf{f}_v))$ as defined in Eq. (9).

**Solver and numerical stability:** Our formulation defined in Eq. (9) is differentiable on the whole domain $\mathcal{F}$. Moreover it does not require reprojection on the hyperbolic domain due to the mapping $g_\beta : \mathcal{F} \to \mathcal{H}^{d,\beta}$. We do not need to use Riemannian stochastic gradient descent (RSGD) as in (Nickel & Kiela, 2018) and can thus use any momentum-based method that keeps track of the gradient history. In our experiments we use standard SGD optimization with momentum.

## 4 Results

We evaluate the Lorentzian distance in three different tasks. The first one considers similarity constraints based on subsumption in hierarchies as described in Section 3.3. The second experiment performs binary classification to determine whether or not a test node belongs to a specific subtree of the hierarchy. The final experiment is qualitative in nature and presents a community detection algorithm based on hierarchical clustering with hyperbolic distances.

### 4.1 Representation of hierarchies

The goal of the first experiment is to learn the hyperbolic representation of a hierarchy. To this end, we consider the same evaluation protocol as Nickel & Kiela (2018) and the same datasets. Each dataset can be seen as a Directed Acyclic Graph (DAG) where a parent node links to its children and descendants in the tree to create a set of edges $\mathcal{D}$ described in Section 3.3. The quality of the learned embeddings is measured with the following metrics: the Mean Rank (MR), the Mean Average Precision (MAP) and the Spearman rank-order correlation $p$ (SROC) between the Euclidean norms of the learned embeddings and the normalized ranks of nodes. Following (Nickel & Kiela, 2018), the normalized rank of a node $u$ is formulated:

$$\mathrm{rank}(u) = \frac{\mathrm{sp}(u)}{\mathrm{sp}(u) + \mathrm{lp}(u)} \in [0, 1] \tag{13}$$

where $\mathrm{lp}(u)$ is the longest path between $u$ and one of its descendant leaves; $\mathrm{sp}(u)$ is the shortest path from the root of the hierarchy to $u$. A leaf in the hierarchy tree then has a normalized rank equal to 1, and nodes closer to the root have smaller normalized rank. $\rho$ is measured as the SROC between the list of Euclidean norms of embeddings and their normalized rank.

Table 1: Evaluation of Taxonomy embeddings. MR: Mean Rank (lower is better). MAP: Mean Average Precision (higher is better). $\rho$: Spearman's rank-order correlation (higher is better).

| Method | | $d_{\mathcal{P}}$ in $\mathcal{P}^d$ | $d_{\mathcal{P}}$ in $\mathcal{H}^d$ | Ours $\beta = 0.01$ $\lambda = 0$ | Ours $\beta = 0.1$ $\lambda = 0$ | Ours $\beta = 1$ $\lambda = 0$ | Ours $\beta = 0.01$ $\lambda = 0.01$ |
|---|---|---|---|---|---|---|---|
| | MR | 4.02 | 2.95 | **1.46** | 1.59 | 1.72 | 1.47 |
| WordNet Nouns | MAP | 86.5 | 92.8 | 94.0 | 93.5 | 91.5 | **94.7** |
| | $\rho$ | 58.5 | 59.5 | 40.2 | 45.2 | 43.1 | **71.1** |
| | MR | 1.35 | 1.23 | **1.11** | 1.14 | 1.23 | 1.13 |
| WordNet Verbs | MAP | 91.2 | 93.5 | **94.6** | 93.7 | 91.9 | 94.0 |
| | $\rho$ | 55.1 | 56.6 | 36.8 | 38.7 | 37.2 | **73.0** |
| | MR | 1.23 | 1.17 | **1.06** | **1.06** | 1.09 | **1.06** |
| EuroVoc | MAP | 94.4 | **96.5** | **96.5** | 96.0 | 95.0 | 96.1 |
| | $\rho$ | 61.4 | **67.5** | 41.8 | 44.2 | 45.6 | 61.7 |
| | MR | 1.71 | 1.63 | **1.03** | 1.06 | 1.16 | 1.04 |
| ACM | MAP | 94.8 | 97.0 | **98.8** | 96.9 | 94.1 | 98.1 |
| | $\rho$ | 62.9 | 65.9 | 53.9 | 55.9 | 46.7 | **66.4** |
| | MR | 12.8 | 12.4 | 1.31 | **1.30** | 1.40 | 1.33 |
| MeSH | MAP | 79.4 | 79.9 | 90.1 | **90.5** | 85.5 | 90.3 |
| | $\rho$ | 74.9 | 76.3 | 46.1 | 47.2 | 41.5 | **78.7** |

Table 2: Percentage of node representations that have smaller Euclidean norm than their children in the tree when $\lambda = 0$

| Method | Wordnet Nouns | Wordnet Verbs | EuroVoc | ACM | MeSH |
|---|---|---|---|---|---|
| Ours ($\beta = 0.01$) | 97.2% | 96.3% | 98.6% | 98.3% | 97.7% |
| Ours ($\beta = 0.1$) | 97.9% | 96.0% | 98.4% | 98.0% | 97.5% |
| Ours ($\beta = 1$) | 94.3% | 92.0% | 98.3% | 94.3% | 93.2% |

We learn the representations $\{\mathbf{f}_i \in \mathcal{F}^d\}_{i=1}^n$ that minimize the following problem:

$$L_{neighborhood} + \lambda \sum_{\{(u,v):\text{rank}(u)<\text{rank}(v)\}} \max(\|\mathbf{f}_u\|^2 - \|\mathbf{f}_v\|^2, 0) \qquad (14)$$

where $\lambda \geq 0$ is a regularization parameter. The first term defined in Eq. (12) tries to get similar examples close to each other. The second term tries to satisfy the order of the embedding norms to match the normalized ranks. Using Theorem 3.1, we optimize Euclidean norms in $\mathcal{F}^d$.

**Datasets:** We consider the following datasets: (1) *2012 ACM Computing Classification System:* is classification system for the computing field used by ACM journal. (2) *EuroVoc:* is a thesaurus maintained by the European Union. (3) *Medical Subject Headings (MeSH):* (Rogers, 1963) is a medical thesaurus provided by the U.S. National Library of Medicine. (4) *Wordnet:* (Miller, 1998) is a large lexical database. As in Nickel & Kiela (2018), we consider the noun and verb hierarchy of WordNet. More details about these datasets can be found in (Nickel & Kiela, 2018).

**Implementation details:** Following (Nickel & Kiela, 2017)[3], we implemented our method in Pytorch 0.3.1. We use the standard SGD optimizer with a learning rate of 0.1 and momentum of 0.9. For the largest datasets *Wordnet Nouns* and *MeSH*, we stop training after 1500 epochs. We stop training at 3000 epochs for the other datasets. The mini-batch size is 50, and the number of sampled negatives per example is 50. The weights of the embeddings are initialized from the continuous uniform distribution in the interval $[-10^{-4}, 10^{-4}]$. The dimensionality of our embeddings is 10. To sample $\{(u,v) : \text{rank}(u) < \text{rank}(v)\}$ in a mini-batch, we randomly sample $\kappa$ examples from the set of positive and negative examples that are sampled for $L_{neighborhood}$, we then select 5% of the possible

---

[3]We use the source code available at `https://github.com/facebookresearch/poincare-embeddings`

ordered pairs. $\kappa = 150$ for all the datasets, except for WordNet nouns where $\kappa = 50$ and MeSH where $\kappa = 100$ due to their large size.

**Results:** We compare in Table 1 the Poincaré distance metric as optimized in (Nickel & Kiela, 2017; 2018) with our method for different values of $\beta$ and $\lambda$ (indicated in the table). Here we separately analyze the case where $\lambda = 0$, which corresponds to using the same constraints as those reported in (Nickel & Kiela, 2018), and where $\lambda > 0$.

- Case where $\lambda = 0$: Our approach obtains better Mean Rank and Mean Average Precision scores than Nickel & Kiela (2018) for small values of $\beta$ when we use the same constraints. The fact that the retrieval performance of our approach changes with different values of $\beta \in \{0.01, 0.1, 1\}$ shows that the curvature of the space has an impact on the distances between examples and on the behavior of the model. As explained in Section 3.2, the squared Lorentzian distance tends to behave more radially (*i.e.* the distance tends to decrease along the ray that can be written $\tau \boldsymbol{\mu}$ where $\tau \geq 0$ and $\boldsymbol{\mu}$ is the centroid) as $\beta$ decreases. Children then tend to have larger Euclidean norm than their parents while being close *w.r.t.* the Lorentzian distance. We evaluate in Table 2 the percentage of nodes that have a Euclidean norm greater than their parent in the tree. More than $90\%$ of pairs (parent,child) satisfy the desired order of Euclidean norms. The percentage increases with smaller values of $\beta$, this illustrates our point on the impact on the Euclidean norm of the center of mass.

On the other hand, our approach obtains worse performance for the SROC metric which evaluates how the order of the Euclidean norms is correlated with their normalized rank/depth in the hierarchy tree. This result is expected due to the formulation of the constraints of $L_{neighborhood}$ that considers only local similarity orders between pairs of examples. The loss $L_{neighborhood}$ does not take into account the global structure of the tree; it only considers whether pairs of concepts subsume each other or not. As a consequence, although the root of a subtree may have a smaller Euclidean norm than all the other nodes of the subtree, this does not necessarily enforce its children to have comparable norms. They can have Euclidean norms smaller or greater than the norms of the descendants of their siblings. In other words, although our representations do not preserve the global structure of the tree, most subtrees preserve produce a root with smaller Euclidean norm than their descendants.

- Case where $\lambda > 0$: As a consequence of the worse SROC performance, we evaluate the performance of our model when including normalized rank information during training. As can be seen in Table 1, this greatly improves the SROC performance and outperforms the baselines (Nickel & Kiela, 2017; 2018) for all the evaluation metrics on some datasets. The Mean Rank and Mean Average Precision performances remain comparable with the case where $\lambda = 0$. Global rank information can then be used during training without having a significant impact on retrieval performance.

In conclusion, we have shown that the curvature of the hyperbolic space has an impact on the retrieval performance of the model. Moreover, the fact the Euclidean norms of parents tend to be smaller than those of children as $\beta > 0$ decreases experimentally demonstrates that the most similar points (*i.e.* high level nodes) tend to get closer to the origin as their set of similar points (*i.e.* number of descendant nodes) increases.

## 4.2 BINARY CLASSIFICATION

Another task of interest for hyperbolic representations is to determine whether a given node belongs to a specific subtree of the hierarchy or not. We follow the same binary classification protocol as Ganea et al. (2018) on the same datasets. We describe their protocol below.

Ganea et al. (2018) extract some pre-trained hyperbolic embeddings of the WordNet nouns hierarchy, those representations are learned with the Poincaré metric. They then consider four subtrees whose roots are the following synsets: animal.n.01, group.n.01, worker.n.01 and mammal.n.01. For each subtree, they consider that every node that belongs to it is positive and all the other nodes of Wordnet nouns are negative. They then select $80\%$ of the positive nodes for training, the rest for test. They select the same percentage of negative nodes for training and test. At test time, they evaluate the F1 score of the binary classification performance. They do it for 3 different training/test splits. We refer to Ganea et al. (2018) for details on the baselines.

Our goal is to evaluate the relevance of the Lorentzian distance to represent hierarchical datasets. We then use the embeddings trained with our approach (with $\beta = 0.01$ and different values of $\lambda$) and classify a test node by assigning it to the category of the nearest training example *w.r.t.* the Lorentzian

Table 3: Test F1 classification scores for four different subtrees of WordNet noun tree.

| Dataset | animal.n.01 | group.n.01 | worker.n.01 | mammal.n.01 |
|---|---|---|---|---|
| (Ganea et al., 2018) | $99.26 \pm 0.59\%$ | $91.91 \pm 3.07\%$ | $66.83 \pm 11.83\%$ | $91.37 \pm 6.09\%$ |
| Euclidean dist | $99.36 \pm 0.18\%$ | $91.38 \pm 1.19\%$ | $47.29 \pm 3.93\%$ | $77.76 \pm 5.08\%$ |
| $\log_0$ + Eucl | $98.27 \pm 0.70\%$ | $91.41 \pm 0.18\%$ | $36.66 \pm 2.74\%$ | $56.11 \pm 2.21\%$ |
| Ours ($\beta = \lambda = 0.01$) | $99.57 \pm 0.24\%$ | $99.75 \pm 0.11\%$ | $94.50 \pm 1.21\%$ | $96.65 \pm 1.18\%$ |
| Ours ($\beta = 0.01, \lambda = 0$) | $99.77 \pm 0.17\%$ | $99.86 \pm 0.03\%$ | $96.32 \pm 1.05\%$ | $97.73 \pm 0.86\%$ |

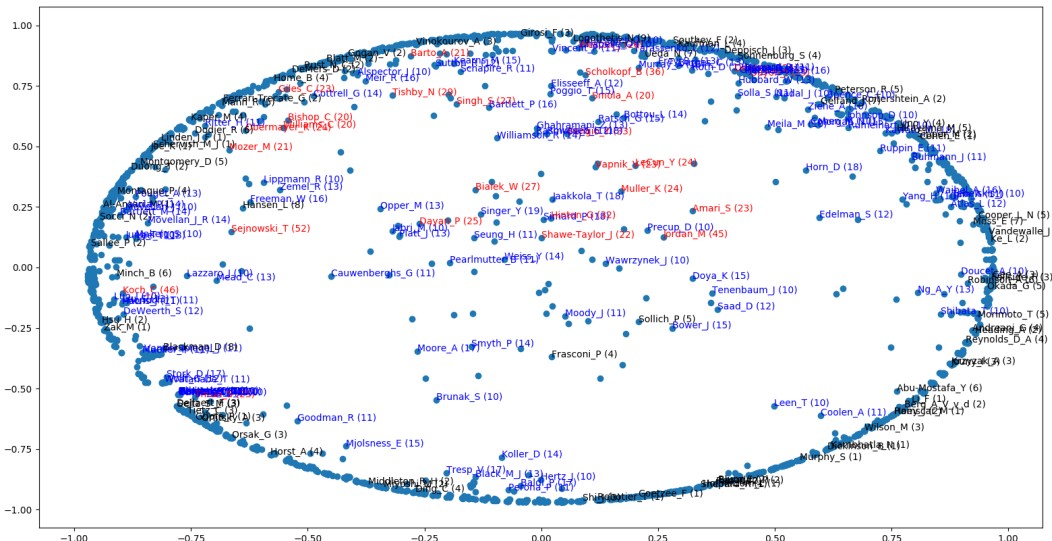

Figure 2: 2-dimensional Poincaré representations learned when optimizing Eq. (14) with $\lambda = 0.01$

distance (with $\beta = 1$). The test performance of our approach is reported in Table 3. It outperforms the classification performance of the baselines, which shows that our embeddings can also be used to perform classification.

## 4.3 COMMUNITY DETECTION

In our last experiment, our dataset does not correspond to a hierarchy tree. It is an undirected graph $G = (\mathcal{V}, \mathcal{D})$ where $\mathcal{V}$ is the set of vertices/nodes, and $\mathcal{D}$ the set of edges. We use the co-authorship information of the NIPS 1-17 dataset (Globerson et al., 2007) which contains information from papers published at NIPS from 1988 to 2003.

The graph $G$ is constructed by considering each author as a node, and an edge $(u, v) \in \mathcal{D}$ is created iff the authors $u \in \mathcal{V}$ and $v \in \mathcal{D}$ are co-authors. The number of authors is 2865 and the number of edges is $|\mathcal{D}| = 9466$. Only $|\mathcal{V}| = 2715$ authors have at least one co-author and the edges are neither weighted by the number of authors per paper nor by the number of times a pair of authors are co-authors.

The main difference with the hierarchical case is that the transitivity relation $(u, v) \in \mathcal{D}, (v, w) \in \mathcal{D} \Rightarrow (u, w) \in \mathcal{D}$ is not necessarily true in the undirected graph case.

Figures 2 and 4 illustrate the 2-dimensional Poincaré representations learned with Eq. (14) for different values of $\lambda \geq 0$. The number of co-authors is written in parenthesis, authors that have at least 20 co-authors are in red, those between 10 and 19 co-authors in blue. The MR and MAP are similar between both approaches (the MR is about 20 and the MAP about $55\%$) but the SROC with the number of co-authors is $93\%$ for Fig. 2. One can observe some communities such as the kernel method researchers (Shawe-Taylor, Vapnik, Smola etc...) are close to the same radius.

Table 4: Examples of clusters extracted from the hierarchical complete-linkage clustering algorithm

| | |
|---|---|
| Computer vision (California) | Malik J, Belongie S, Fowlkes C, Martin D, Torresani L, Hertzmann A, Bregler C, Omohundro S, Stolcke A, Allinson N, Moon K |
| Machine learning (MIT & California) | Jordan M, Tenenbaum J, Russell S, El Ghaoui L, Ng A, Blei D Bhattacharyya C, Nilim A, Lanckriet G, Xing E |
| Kernel methods (cluster 1) | Scholkopf B, Smola A, Weston J, Bousquet O, Chapelle O, Gretton A |
| Kernel methods (cluster 2) | Shawe-Taylor J, Cristianini N, Platt J, Campbell C |

When learning 10-dimensional representations, the MR is 1.0 and the MAP is $100\%$. When $\lambda = 0$, the embeddings with smallest Euclidean norm are in that order: Sejnowski T. (52), Jordan M. (45), Tishby N. (29), Dayan P. (25), Koch C. (46), Singh S. (27), Hinton G. (32), Scholkopf B. (36), Mozer M. (21), Bialek W. (27), Singer Y. (19), Vapnik V. (23), Bengio Y. (33), Shawe-Taylor J. (22), Zemel R. (13). The authors with a large number of co-authors then tend to have smallest Euclidean norms.

Since hyperbolic distances are appropriate to represent hierarchies, we perform a hierarchical agglomerative clustering with the learned squared Lorentzian distance (with dimensionality $d = 10$) based on complete-linkage clustering (Defays, 1977) (*i.e.* linkage uses the maximum distances between all observations of two sets to merge them into a same cluster). Some examples of extracted clusters are given in Table 4. Fig 5 illustrates how the researchers from the kernel method clusters of Table 4 are represented in Fig 2.

Most of them are close to the same radius of the Poincaré ball. This simple example shows that the Lorentzian distance can be applied to datasets that are not only hierarchical.

## 5 CONCLUSION

In this paper, we proposed a distance learning approach based on the Lorentzian distance to represent hierarchically-structured datasets. Unlike most of the literature that considers the unit hyperboloid model, we show that the performance of the learned model can be improved when the chosen hyperboloid model has low curvature. We give a formulation of the centroid *w.r.t.* the squared Lorentzian distance as a function of the curvature, and we show that the Euclidean norm of its projection in the Poincaré ball decreases as the curvature decreases. Hierarchy constraints are generally formulated such that high-level nodes are similar to all their descendants and thus their representation should be close to the centroid of the descendants. Using low curvature implicitly enforces high-level nodes to have smaller Euclidean norm than their descendants and is therefore is more appropriate for learning representations of hierachically-structured datasets.

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

# A PROOFS

## A.1 PROOF OF THEOREM 3.1:

• We first show ($\Rightarrow$): let us note $\alpha = \|\mathbf{a}\|^2$, we have the following equality $\|h(g_\beta(\mathbf{a}))\|^2 = \frac{\alpha}{(\sqrt{\alpha+1}+1)^2}$ which is an increasing function in $\alpha$ when $\alpha$ is positive.

• We now show ($\Leftarrow$): We give the formulation of $g_\beta^{-1} \circ h^{-1}$:

$$\forall \mathbf{c} = (c_1, \cdots, c_d) \in \mathcal{P}^d, h^{-1}(\mathbf{c}) = \left( \sqrt{\left\| \frac{2\mathbf{c}}{1 - \|\mathbf{c}\|^2} \right\|^2 + \beta}, \frac{2c_1}{1 - \|\mathbf{c}\|^2}, \cdots, \frac{2c_d}{1 - \|\mathbf{c}\|^2} \right) \in \mathcal{H}^{d,\beta} \tag{15}$$

$$= \left( \sqrt{\left\| \frac{2\mathbf{c}}{1 - \|\mathbf{c}\|^2} \right\|^2 + \beta}, \frac{2\mathbf{c}}{1 - \|\mathbf{c}\|^2} \right) \in \mathcal{H}^{d,\beta} \tag{16}$$

The vector $g_\beta^{-1}(h^{-1}(\mathbf{c}))$ is obtained by removing the first element of $h^{-1}(\mathbf{c})$:

$$\forall \mathbf{c} \in \mathcal{P}^d, g_\beta^{-1}(h^{-1}(\mathbf{c})) = \frac{2\mathbf{c}}{1 - \|\mathbf{c}\|^2} \tag{17}$$

By using these definitions, we have the following implication: we recall that $\forall \mathbf{c} \in \mathcal{P}^d, \|\mathbf{c}\| < 1$ due to the definition of $\mathcal{P}^d$. Let us note $\gamma = \|\mathbf{c}\|^2 < 1$, we have $\|g_\beta^{-1}(h^{-1}(\mathbf{c}))\|^2 = \frac{4\gamma}{(1-\gamma)^2}$ which is an increasing function in $\gamma$ on $(0, 1)$. QED

## A.2 PROOF OF LEMMA 3.2

By using the linearity of the Lorentzian inner product, we have the following equality:

$$\max_{\boldsymbol{\mu} \in \mathcal{H}^{d,\beta}} \sum_{i=1}^n \nu_i \langle \mathbf{x}_i, \boldsymbol{\mu} \rangle_{\mathcal{L}} = \max_{\boldsymbol{\mu} \in \mathcal{H}^{d,\beta}} \langle \sum_{i=1}^n \nu_i \mathbf{x}_i, \boldsymbol{\mu} \rangle_{\mathcal{L}} \tag{18}$$

Since $\boldsymbol{\mu} \in \mathcal{H}^{d,\beta}$ and $\forall \mathbf{a} \in \mathcal{H}^{d,\beta}, \mathbf{b} \in \mathcal{H}^{d,\beta}, \langle \mathbf{a}, \mathbf{b} \rangle_{\mathcal{L}}$ is maximized if $\mathbf{a} = \mathbf{b}$, Eq. (18) is then maximized for the scaling factor $\gamma > 0$ that satisfies $(\gamma \sum_{i=1}^n \nu_i \mathbf{x}_i) = \boldsymbol{\mu} \in \mathcal{H}^{d,\beta}$. We explain in the next paragraph how to construct such a $\gamma$.

For any positive time-like vector $\mathbf{a}$, the vector $\mathbf{b} = \frac{1}{\|\|\mathbf{a}\|_{\mathcal{L}}\|} \mathbf{a}$ satisfies $\langle \mathbf{b}, \mathbf{b} \rangle_{\mathcal{L}} = \frac{\langle \mathbf{a}, \mathbf{a} \rangle_{\mathcal{L}}}{|\langle \mathbf{a}, \mathbf{a} \rangle_{\mathcal{L}}|} = -1$. Therefore, $\beta \langle \mathbf{b}, \mathbf{b} \rangle_{\mathcal{L}} = \langle \sqrt{\beta}\mathbf{b}, \sqrt{\beta}\mathbf{b} \rangle_{\mathcal{L}} = -\beta$ (i.e. $\sqrt{\beta}\mathbf{b} \in \mathcal{H}^{d,\beta}$ by definition since $\sqrt{\beta}\mathbf{b}$ is positive) and we then have $\gamma = \frac{\sqrt{\beta}}{\|\|\sum_{i=1}^n \nu_i \mathbf{x}_i\|_{\mathcal{L}}\|}$. QED

## A.3 PROOF OF THEOREM 3.4

Let $\mathbf{f}_1, \cdots, \mathbf{f}_n$ be a set of at least two different points in $\mathcal{F}^d$, and $\boldsymbol{\mu}$ be the centroid of the set $g_\beta(\mathbf{f}_1), \cdots, g_\beta(\mathbf{f}_n)$ (with same weighting $\forall i, \nu_i = 1$). The squared Euclidean norm of $g_\beta^{-1}(\boldsymbol{\mu}) \in \mathcal{F}^d$ is:

$$\|g_\beta^{-1}(\boldsymbol{\mu})\|^2 = \frac{\beta}{-\|\sum_{i=1}^n g_\beta(\mathbf{f}_i)\|_{\mathcal{L}}^2} \|\sum_{i=1}^n \mathbf{f}_i\|^2 \tag{19}$$

Since $\|\sum_{i=1}^n \mathbf{f}_i\|^2$ does not depend on $\beta$ and the factor $\frac{\beta}{-\|\sum_{i=1}^n g_\beta(\mathbf{f}_i)\|_{\mathcal{L}}^2}$ is positive for $\beta > 0$, we only need to show that it is a increasing function in $\beta$, or equivalently, that its reciprocal is decreasing in $\beta$.

The reciprocal of $\frac{\beta}{-\|\sum_{i=1}^{n} g_\beta(\mathbf{f}_i)\|_{\mathcal{L}}^2}$ is:

$$\frac{-\|\sum_{i=1}^{n} g_\beta(\mathbf{f}_i)\|_{\mathcal{L}}^2}{\beta} = \frac{-[\sum_{i=1}^{n} \|g_\beta(\mathbf{f}_i)\|_{\mathcal{L}}^2 + 2\sum_{i=1}^{n}\sum_{j>i}^{n}\langle g_\beta(\mathbf{f}_i), g_\beta(\mathbf{f}_j)\rangle_{\mathcal{L}}]}{\beta} \tag{20}$$

$$= n - \frac{2}{\beta}\sum_{i=1}^{n}\sum_{j>i}^{n}\langle g_\beta(\mathbf{f}_i), g_\beta(\mathbf{f}_j)\rangle_{\mathcal{L}} \tag{21}$$

$$= n + \frac{2}{\beta}\sum_{i=1}^{n}\sum_{j>i}^{n}\langle \sqrt{\|\mathbf{f}_i\|^2 + \beta}, \sqrt{\|\mathbf{f}_j\|^2 + \beta}\rangle - \langle \mathbf{f}_i, \mathbf{f}_j\rangle \tag{22}$$

We then need to show that the function

$$\frac{1}{\beta}\left[\langle\sqrt{\|\mathbf{f}_i\|^2 + \beta}, \sqrt{\|\mathbf{f}_j\|^2 + \beta}\rangle - \langle \mathbf{f}_i, \mathbf{f}_j\rangle\right] \tag{23}$$

is decreasing in $\beta$ if $\mathbf{f}_i \neq \mathbf{f}_j$. For this purpose, We study the sign of its gradient. The gradient of Eq. (23) *w.r.t.* $\beta$ is written:

$$\frac{2\langle\mathbf{f}_i, \mathbf{f}_j\rangle\sqrt{\|\mathbf{f}_i\|^2 + \beta}\sqrt{\|\mathbf{f}_j\|^2 + \beta} - \beta(\|\mathbf{f}_i\|^2 + \|\mathbf{f}_j\|^2) - 2\|\mathbf{f}_i\|^2\|\mathbf{f}_j\|^2}{2\beta^2\sqrt{\|\mathbf{f}_i\|^2 + \beta}\sqrt{\|\mathbf{f}_j\|^2 + \beta}} \tag{24}$$

The denominator is positive. If $\langle\mathbf{f}_i, \mathbf{f}_j\rangle$ is negative, then Eq. (24) is negative. If $\langle\mathbf{f}_i, \mathbf{f}_j\rangle = 0$, then Eq. (24) is 0 if $\mathbf{f}_i = \mathbf{f}_j = \mathbf{0}$, and negative otherwise. Otherwise, the Cauchy-Schwarz inequality is used to prove that Eq. (24) is negative. In other words, we need to prove that (assuming that $\langle\mathbf{f}_i, \mathbf{f}_j\rangle$ is positive):

$$2\langle\mathbf{f}_i, \mathbf{f}_j\rangle\sqrt{\|\mathbf{f}_i\|^2 + \beta}\sqrt{\|\mathbf{f}_j\|^2 + \beta} - \beta(\|\mathbf{f}_i\|^2 + \|\mathbf{f}_j\|^2) - 2\|\mathbf{f}_i\|^2\|\mathbf{f}_j\|^2 < 0 \tag{25}$$

$$\Longleftrightarrow 2\langle\mathbf{f}_i, \mathbf{f}_j\rangle\sqrt{\|\mathbf{f}_i\|^2 + \beta}\sqrt{\|\mathbf{f}_j\|^2 + \beta} < \beta(\|\mathbf{f}_i\|^2 + \|\mathbf{f}_j\|^2) + 2\|\mathbf{f}_i\|^2\|\mathbf{f}_j\|^2 \tag{26}$$

$$\Longleftrightarrow (2\langle\mathbf{f}_i, \mathbf{f}_j\rangle\sqrt{\|\mathbf{f}_i\|^2 + \beta}\sqrt{\|\mathbf{f}_j\|^2 + \beta})^2 < [\beta(\|\mathbf{f}_i\|^2 + \|\mathbf{f}_j\|^2) + 2\|\mathbf{f}_i\|^2\|\mathbf{f}_j\|^2]^2 \tag{27}$$

$$\Longleftrightarrow 4\langle\mathbf{f}_i, \mathbf{f}_j\rangle^2(\|\mathbf{f}_i\|^2 + \beta)(\|\mathbf{f}_j\|^2 + \beta) = 4\langle\mathbf{f}_i, \mathbf{f}_j\rangle^2(\|\mathbf{f}_i\|^2\|\mathbf{f}_j\|^2 + \beta\|\mathbf{f}_i\|^2 + \beta\|\mathbf{f}_j\|^2 + \beta^2) \tag{28}$$

$$< 4\|\mathbf{f}_i\|^4\|\mathbf{f}_j\|^4 + 4\beta\|\mathbf{f}_i\|^2\|\mathbf{f}_j\|^2(\|\mathbf{f}_i\|^2 + \|\mathbf{f}_j\|^2) + \beta^2\|\mathbf{f}_i\|^4 + \beta^2\|\mathbf{f}_j\|^4 + 2\beta^2\|\mathbf{f}_i\|^2\|\mathbf{f}_j\|^2 \tag{29}$$

Since, by the Cauchy-Schwarz inequality, we have:

$$\langle\mathbf{f}_i, \mathbf{f}_j\rangle^2 \leq \|\mathbf{f}_i\|^2\|\mathbf{f}_j\|^2 \tag{30}$$

we then have (term by term):

$$4\langle\mathbf{f}_i, \mathbf{f}_j\rangle^2\|\mathbf{f}_i\|^2\|\mathbf{f}_j\|^2 \leq 4\|\mathbf{f}_i\|^4\|\mathbf{f}_j\|^4 \tag{31}$$

$$4\beta\langle\mathbf{f}_i, \mathbf{f}_j\rangle^2\|\mathbf{f}_i\|^2 \leq 4\beta\|\mathbf{f}_i\|^2\|\mathbf{f}_j\|^2\|\mathbf{f}_i\|^2 \tag{32}$$

$$4\beta\langle\mathbf{f}_i, \mathbf{f}_j\rangle^2\|\mathbf{f}_j\|^2 \leq 4\beta\|\mathbf{f}_i\|^2\|\mathbf{f}_j\|^2\|\mathbf{f}_j\|^2 \tag{33}$$

$$2\beta^2\langle\mathbf{f}_i, \mathbf{f}_j\rangle^2 \leq 2\beta^2\|\mathbf{f}_i\|^2\|\mathbf{f}_j\|^2 \tag{34}$$

$$2\beta^2\langle\mathbf{f}_i, \mathbf{f}_j\rangle^2 \leq 2\beta^2\|\mathbf{f}_i\|^2\|\mathbf{f}_j\|^2 \leq \beta^2\|\mathbf{f}_i\|^4 + \beta^2\|\mathbf{f}_j\|^4 \tag{35}$$

Eq. (35) is explained by $\beta^2\|\mathbf{f}_i\|^4 + \beta^2\|\mathbf{f}_j\|^4 - 2\beta^2\|\mathbf{f}_i\|^2\|\mathbf{f}_j\|^2 = \beta^2(\|\mathbf{f}_i\|^2 - \|\mathbf{f}_j\|^2)^2 \geq 0$.

Eq. (35) is then an equality iff $\|\mathbf{f}_i\|^2 = \|\mathbf{f}_j\|^2$, and the Cauchy-Schwarz relation is an equality iff $\mathbf{f}_i$ are linearly $\mathbf{f}_j$ dependent. Both of these conditions are satisfied iff $\mathbf{f}_i = \mathbf{f}_j$ (assuming that $\langle\mathbf{f}_i, \mathbf{f}_j\rangle$ is positive). However, we assume that there are at least two different points $\mathbf{f}_i$ and $\mathbf{f}_j$ such that $\mathbf{f}_i \neq \mathbf{f}_j$, which then implies that Eq. (24) is negative for this choice of $\mathbf{f}_i$ and $\mathbf{f}_j$.

This proves that the Euclidean norm in $\mathcal{F}^d$ of the centroid of different points decreases as $\beta > 0$ decreases. From Section A.1, the Euclidean norm of a point in $\mathcal{P}^d$ increases as the Euclidean norm of its projection in $\mathcal{F}^d$ increases, which completes the proof.

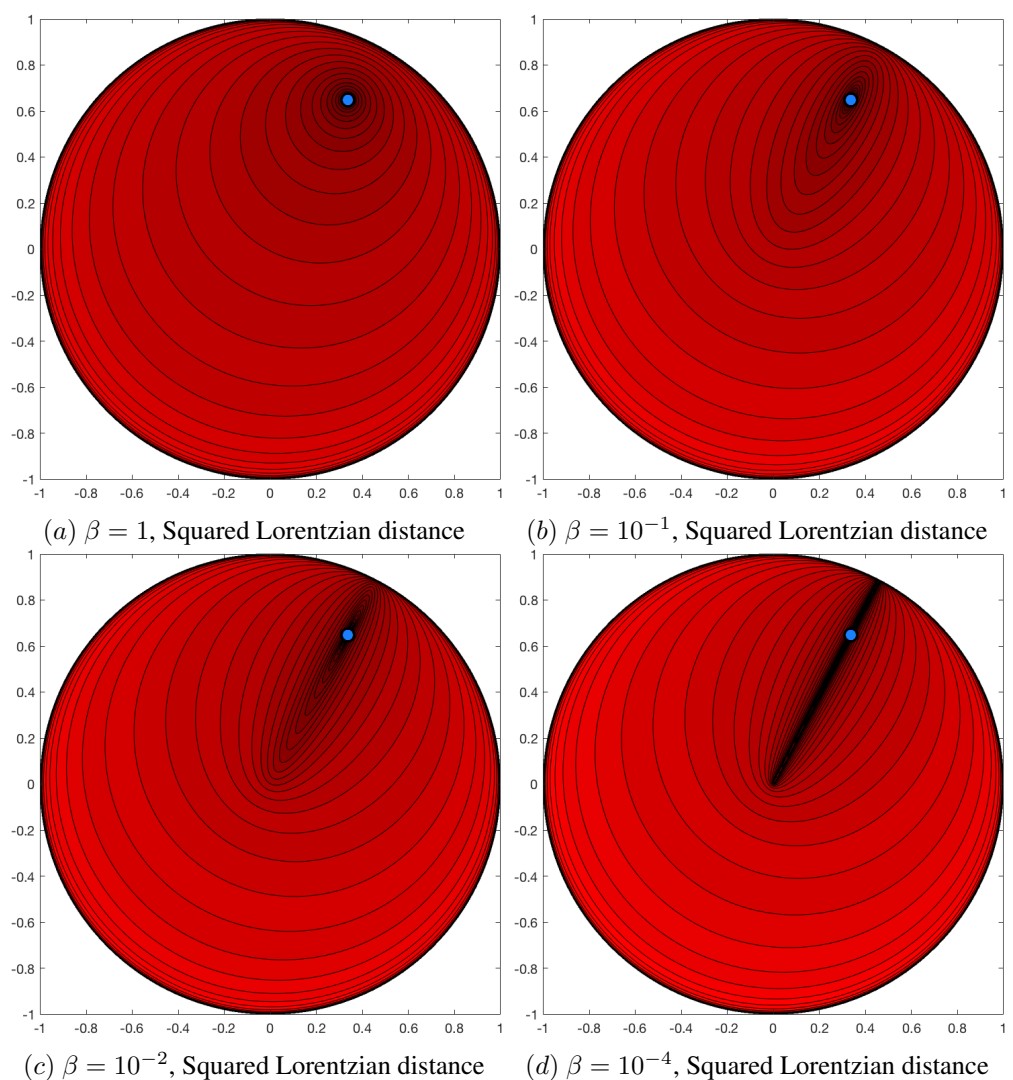

(a) $\beta = 1$, Squared Lorentzian distance

(b) $\beta = 10^{-1}$, Squared Lorentzian distance

(c) $\beta = 10^{-2}$, Squared Lorentzian distance

(d) $\beta = 10^{-4}$, Squared Lorentzian distance

Figure 3: Level set of distances to the point in purple depending on the parameter $\beta$. The distance tends to be smaller along the radius that contains the point as explained in Section 3.2.

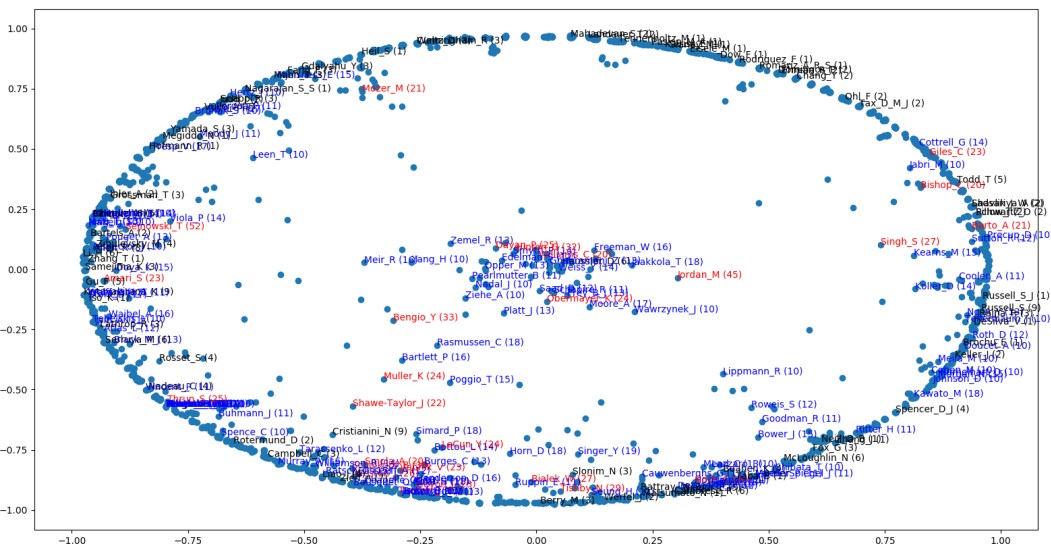

Figure 4: 2-dimensional Poincaré representations learned when optimizing Eq. (14) with $\lambda = 0$

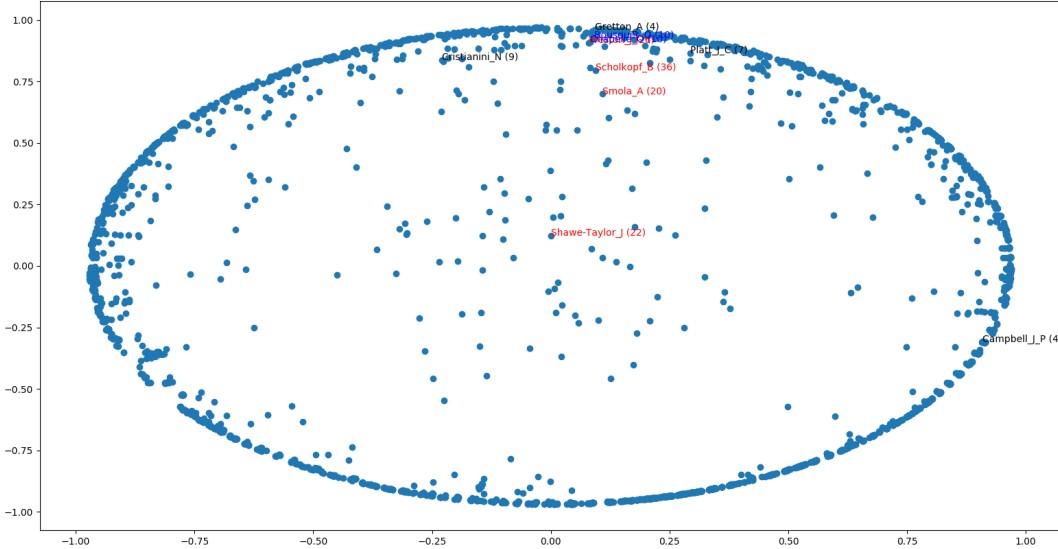

Figure 5: 2-dimensional Poincaré representations of researchers reported in the two kernel method clusters of Table 4, as in Fig 2, they are learned when optimizing Eq. (14) with $\lambda = 0$

