# OpenReview forum: "Lorentzian Distance Learning"
_ICLR.cc/2019/Conference_

### Official Review · AnonReviewer2 · 2018-10-29
**An incremental work on hyperbolic embedding with Lorentzian Distance**

**Rating:** 5
**Confidence:** 4

**Review:**

This paper proposed an unsupervised embedding method for hierarchical or graph datasets. The embedding space is a hyperbolic space as in several recent works such as (Nickel & Kiela, 2017). The author(s) showed that using the proposed embedding, the optimization has better numerical stability and better performance.

I am convinced of the correctness and the experiment results, and I appreciate that the paper is well written with interesting interpretations such as the demonstration of the centroid. However, the novelty of this contribution is limited and may not meet the publication standard of ICLR.  I suggest that the authors enhance the results and resubmit this work in a future venue.

Theoretically, there are three theorems in section 3:

Theorem 3.1 shows the coordinate transformation from the proposed parametrization to hyperboloid then to Poincare ball preserves the monotonicity of the Euclidean norm. This is straightforward by writing down the two transformations.

Lemma 3.2 and theorem 3.3 state the centroid in closed expression based on the Lorentzian distance, taking advantage that the  Lorentzian distance is in a bi-linear form (no need to take the arcosh()  therefore the analysis are much more simplified) These results are quite striaghtforward from the expression of the energy function.

Theorem 3.4  (centroid computation with constraints) shows that minimizing the energy function
$\sum_{i} d_L^2 (x_i, a)$, when a is constrained to a discrete set, is equivalent to minimizing $d_L(c,a)$, where $c$ is given by Lemma 3.2.
This is interesting as compared to the previous two theorems, but it is not clear whether/how this equivalence is used in the proposed embedding.

Technically, there are three novel contributions,

1. The proposed unconstrained reparametrization of the hyperboloid model does not require to project the embedding points onto the hyperbolic manifold in each update.

2. The cost is based on the Lorentzian Distance, that is a monotonic transformation of the Riemannian distance (without taking the arccosh function). Therefore the similarity (a heat kernel applied on the modified distance function) is measured differently than the other works. Informally one can think it as t-SNE v.s. SNE which use different similarity measures in the target embedding.

3. The authors discussed empirically the different choice of beta, which was typically chosen as beta=1 in previous works, showing that tunning the beta hyperparameter can give better embeddings.

These contributions are useful but incremental. Notably, (1) needs more experimental evidence (e.g. a toy example) to show the numerical instability of the other methods, and to show the learning curves of the proposed re-parametrization against the Riemannian stochastic gradient descent, which are not given in the paper.

By listing these theoretical and technical contributions, overall I find that most of these contributions are incremental and not significant enough for ICLR.

---

> ### Author Response · Authors · 2018-11-16
> **Response to AnonReviewer2**
>
> - “Theorem 3.4 (centroid computation with constraints) shows that minimizing the energy function $\sum_{i} d_L^2 (x_i, a)$, when a is constrained to a discrete set, is equivalent to minimizing $d_L(c,a)$, where $c$ is given by Lemma 3.2. This is interesting as compared to the previous two theorems, but it is not clear whether/how this equivalence is used in the proposed embedding.”
>
> We would like to emphasize why this theorem is important for some important contribution of the paper.
> We show that the distances with respect to a set of points can be reduced to calculating the distance to the centroid in the case of the Lorentzian distance. Interpreting distances to a set of points can then be done by interpreting their centroid.
> From this observation, we study the property of the centroid which can be written in closed form. In particular, we show that the Euclidean norm of the centroid decreases as the curvature of the space decreases, representing trees then becomes easier.
>
> - “These contributions are useful but incremental. Notably, (1) needs more experimental evidence (e.g. a toy example) to show the numerical instability of the other methods, and to show the learning curves of the proposed re-parametrization against the Riemannian stochastic gradient descent, which are not given in the paper.”
>
> As mentioned above, our goal is not to prove that our algorithm is more stable than existing approaches although we explain that the Poincaré distance metric is not differentiable everywhere on the domain and its gradient tends to infinity when distances are infinitesimal. Both our approach and the Riemannian stochastic gradient descent are stable, although we can use momentum-based optimizers.
>
> As already mentioned, the main advantage of our approach is that we are able to interpret the behavior of distances with sets of similar points.

---

> > ### Comment · AnonReviewer2 · 2018-12-04
> > **Numerical stability**
> >
> > Thank you very much for the response.
> >
> > I still think there should be some experimental comparison with Riemannian exp map on the advantage of the proposed method from the optimization perspective (e.g. numerical stability as claimed in the paper). This will make the contribution more complete. Notice that the  Poincaré distance metric is differentiable on P^2 (P is the Poincare disk). Why isn't it?

---

> > > ### Author Response · Authors · 2018-12-04
> > > **About the differentiability of the Poincaré metric on P^2**
> > >
> > > The Poincaré metric is not differentiable on P^2 (or P^d in general for any value of d) when the compared points are equal.
> > > The formulation of the Poincaré metric on P^d is given in Eq. (3): when c = d (the optimal case that we might want to obtain for some pairs), Eq. (3) is equal to arcosh(1). However, due to its definition, arcosh(x) = log( x + sqrt( x^2 - 1)) is not differentiable at x = 1 (due to the sqrt term).
> > > We agree that arcosh is differentiable on (1, +inf). Moreover, the points have to be reprojected onto the interior of the unit disk if the optimized variables are in P^2.

---

> > > > ### Comment · AnonReviewer2 · 2018-12-05
> > > > **Reply to differentiability**
> > > >
> > > > Thanks for the reply. I double checked that $d_{\mathcal{P}}(c,d)$ is differentiable on $(c,d)\in(\mathcal{P}^d)^2$. If this is still not clear, please write down your derivations.

---

> > > > > ### Author Response · Authors · 2018-12-06
> > > > > **Reply to differentiability**
> > > > >
> > > > > We thank the reviewer for taking time to check our statement about the differentiability of Eq. (3).
> > > > > For simplicity, we will write the proof in .tex code since we cannot update the pdf file.
> > > > >
> > > > > Let us note $\textbf{c} = (x, c_2)$ and $\textbf{d} = (d_1, d_2)$ the considered points in Eq. (3) when the dimensionality is 2 (as suggested by the reviewer). We show in the following that Eq. (3) is not differentiable when $\textbf{c} = \textbf{d}$. To this end, we consider that our variable is $x$ (i.e. the first element of \textbf{c}).
> > > > > We also note $h = x - d_1$.
> > > > > Note that we could equivalently consider that our variables are the other elements of $\textbf{c}$ or $\textbf{d}$.
> > > > > Since we assume that $\textbf{c} = \textbf{d}$, we have $c_2 = d_2$ and we study the behavior of $x$ when it tends to $d_1$ or equivalently when $h$ tends to 0.
> > > > >
> > > > > When $c_2 = d_2$, Eq. (3) can be written as:
> > > > > \begin{equation}
> > > > > arcosh(f(h)) = log( f(h) + \sqrt{f^2(h) - 1})
> > > > > \end{equation}
> > > > > where
> > > > > \begin{equation}
> > > > > f(h) = 1 + 2 \frac{h^2}{(1 - (d_1 + h)^2 - d_2^2)(1 - d_1^2 - d_2^2)} = 1 + 2 \frac{h^2}{\alpha}
> > > > > \end{equation}
> > > > > where we note $\alpha = (1 - (d_1 + h)^2 - d_2^2)(1 - d_1^2 - d_2^2) > 0$ which is positive since the points $\textbf{c}$ and $\textbf{d}$ are constrained to be in the interior of the unit disk.
> > > > >
> > > > > For all $h$, $arcosh(f(h))$ is then nonnegative.
> > > > >
> > > > > We also have $f(0) = 1$ and $arcosh(f(0)) = 0$.
> > > > >
> > > > > We now study two one-sided limits. The first one is the right-sided limit:
> > > > > \begin{equation}
> > > > > lim_{h \to 0^+} \frac{arcosh(f(h)) - arcosh(f(0))}{h} = lim_{h \to 0^+} \frac{arcosh(f(h))}{h} = lim_{h \to 0^+} (arcosh(f(h)))'
> > > > > \end{equation}
> > > > > which is nonnegative since $arcosh(f(h))$ is nonnegative.
> > > > > The second one is the left-sided limit:
> > > > > \begin{equation}
> > > > > lim_{h \to 0^-} \frac{arcosh(f(h)) - arcosh(f(0))}{h}  = lim_{h \to 0^-} \frac{arcosh(f(h))}{h} = lim_{h \to 0^-} (arcosh(f(h)))'
> > > > > \end{equation}
> > > > > which is nonpositive.
> > > > >
> > > > > We show in the following that these two limits are different, which implies that the function is not differentiable when $\textbf{c} = \textbf{d}$.
> > > > >
> > > > > We then need to show that at least one of these limits is nonzero to show they are different.
> > > > > We study the first one and show that it is nonzero.
> > > > >
> > > > >
> > > > >
> > > > >
> > > > >
> > > > > The derivative of arcosh wrt $z > 1$ is:
> > > > > \begin{equation}
> > > > > arcosh'(z) = \frac{1}{\sqrt{z^2 - 1}}
> > > > > \end{equation}
> > > > >
> > > > > The derivative of $f$ wrt $h$ is:
> > > > > \begin{equation}
> > > > > f'(h) =  \frac{4h (d_1 h + d_2^2 + d_1^2 - 1)}{(d_2^2 + d_1^2 - 1)(h^2 + 2 d_1 h + d_2^2 + d_1^2 - 1)^2}
> > > > > \end{equation}
> > > > >
> > > > > The derivative of $arcosh(f)$ wrt $h$ is then $f'(h) arcosh'(f(h))$ which can be written:
> > > > > \begin{equation}
> > > > > \frac{4h (d_1 h + d_2^2 + d_1^2 - 1)}{(d_2^2 + d_1^2 - 1)(h^2 + 2 d_1 h + d_2^2 + d_1^2 - 1)^2 \sqrt{(1 + 2 \frac{h^2}{(1 - (d_1 + h)^2 - d_2^2)(1 - d_1^2 - d_2^2)})^2-1}}
> > > > > \end{equation}
> > > > > which is equal to:
> > > > > \begin{equation}
> > > > > \frac{4h(d_1 h + d_2^2 + d_1^2 - 1)}{(d_2^2 + d_1^2 - 1)(h^2 + 2 d_1 h + d_2^2 + d_1^2 - 1)^2 \sqrt{\left(2 \frac{h^2}{(1 - (d_1 + h)^2 - d_2^2)(1 - d_1^2 - d_2^2)} \right) \left(2 \frac{h^2}{(1 - (d_1 + h)^2 - d_2^2)(1 - d_1^2 - d_2^2)} + 2\right)}}
> > > > > \end{equation}
> > > > > which is equal to:
> > > > > \begin{equation}
> > > > > \frac{2h(d_1 h + d_2^2 + d_1^2 - 1)}{(h^2 + 2 d_1 h + d_2^2 + d_1^2 - 1) \sqrt{ h^2 \left( h^2 + (1 - (d_1 + h)^2 - d_2^2)(1 - d_1^2 - d_2^2)\right)}}
> > > > > \end{equation}
> > > > > From the above equation, one can see that the right sided limit $L$ is:
> > > > > \begin{equation}
> > > > >     L := lim_{h \to 0^+} (arcosh(f(h)))' = lim_{h \to 0^+} \frac{2(d_1 h + d_2^2 + d_1^2 - 1)}{(h^2 + 2 d_1 h + d_2^2 + d_1^2 - 1) \sqrt{ \left( h^2 + (1 - (d_1 + h)^2 - d_2^2)(1 - d_1^2 - d_2^2)\right)}}
> > > > > \end{equation}
> > > > > which is equal to:
> > > > > \begin{equation}
> > > > > L = \frac{2}{1 - d_2^2 - d_1^2}
> > > > > \end{equation}
> > > > > which is nonzero by definition of the domain of $\textbf{d} = (d_1, d_2)$.
> > > > > For instance, when $\textbf{d} = 0$, we have $L = 2$.
> > > > >
> > > > > A similar proof can show that the left-sided limit is:
> > > > > \begin{equation}
> > > > >     lim_{h \to 0^-} (arcosh(f(h)))' = -L
> > > > > \end{equation}
> > > > >
> > > > > The fact that these two one-sided limits are different shows that Eq. (3) is not differentiable when $\textbf{c} = \textbf{d}$.
> > > > >
> > > > > We would also like to emphasize that L increases as the l2 norm of \textbf{d}$ increases.

---

> > > > > > ### Comment · AnonReviewer2 · 2018-12-08
> > > > > > **The authors are right about their claim on differentiability**
> > > > > >
> > > > > > Thanks for the detailed reply. It was a mistake on my side (I ignored the term x/abs(x) in my derivations). Indeed the proposed parametrization had advantages in its smoothness. However, I still think there should be some numerical evidence on how/whether such an advantage can affect the optimization.
> > > > > >
> > > > > > What do you think? I am open to a discussion on whether this empirical evaluation on numerical stability is necessary, and whether the current experiments are sufficient.

---

### Official Review · AnonReviewer3 · 2018-11-05
**Review - Lorentzian Distance Learning**

**Rating:** 5
**Confidence:** 4

**Review:**

The paper proposes a new approach to compute hyperbolic embeddings based on the squared Lorentzian distance. This choice of distance function is motivated by the observation that the ranking of these distances is equivalent to the ranking of the true hyperbolic distance (e.g., on the hyperboloid). For this reason, the paper proposes to use this distance function in combination with ranking losses as proposed by Nickel & Kiela (2017), as it might be easier to optimize. Moreover, the paper proposes to use Weierstrass coordinates as a representation for points on the hyperboloid.

Hyperbolic embeddings are a promising new research area that fits well into ICLR. Overall, the paper is written well and good to understand. It introduces interesting ideas that are promising to advance hyperbolic embeddings. However, in the current version of the paper, these ideas are not fully developed or their impact is unclear.

For instance, using Weierstrass coordinates as a representations seems to make sense, as it allows to use standard optimization methods without leaving the manifold. However, it is important to note that the optimization is still performed on a Riemannian manifold. For that reason, following the Riemannian gradient along geodesics would still require the exponential map. Moreover, optimization methods like Adam or SVRG are still not directly applicable. Therefore, it seems that the practical benefit of this representation is limited.

Similarly, being able to compute the centroid efficiently in closed form is indeed an interesting aspect of the proposed approach. Moreover, the paper explicitly connects the centroid to the least common ancestor of children in a
tree, what could be very useful to derive new embedding methods. Unfortunately, this is advantage isn't really exploited in the paper. The approach taken in the paper simply uses the loss function of Nickel & Kiela (2017) and this loss doesn't make use of centroids, as the paper notes itself. The only use of the centroid seems then to justify the regularization method, i.e., that parents should have a smaller norm than their children. However, this insight alone seems not particularly novel, as the same insight can be derived for standard hyperbolic method and has, for instance, been discussed in Nickel & Kiela (2017, 2018), Ganea et al (2018), De Sa (2018). Using the centroid to derive new hyperbolic embeddings could be interesting, but, unfortunately, is currently not done in the paper.

Further comments
- p.3: Projection back onto the Poincaré ball/manifold is only necessary when
  the exponential map isn't used. The methods of Nickel & Kiela (2018), Ganea et al (2018) therefore don't have this problem.
- p.7: Since MR and MAP are ranking measures, and the ranking of distances between H^d and the L^2 distance should be identical, it is not clear to me why the experiments show significant differences for these methods when \beta=1
- p.7: Embeddings in the Poincaré ball and the Hyperboloid are both compatible with the regularization method in eq.14 (using their respective norms). It would be interesting to also see results for these methods with regularization.

---

> ### Author Response · Authors · 2018-11-16
> **Response to AnonReviewer3**
>
> - “Using Weierstrass allows to use standard optimization methods without leaving the manifold. However, the optimization is still performed on a Riemannian manifold.”
>
> We understand the concern that a Riemannian optimizer would probably be more appropriate since our representations lie on a manifold.
> However, from Eq. (9), our distance function can also be seen as the sum of a simple bilinear form between real vectors and another term promoting some similarity of their Euclidean norms. This formulation is then very similar to optimizing (squared) Euclidean distances.
>
> - “Computing the centroid in closed form is interesting but isn't really exploited in the paper.“
>
> We agree that we do not explicitly use the closed-form solution of the centroid in our experiments. However, our last theorem explains that minimizing the distances to a set of points is equivalent to minimizing the distance to its centroid.
> Our study of the centroid is important to understand the behavior of our distance function with the considered set of similarity constraints (based on hierarchical relationships).
>
> - “The only use of the centroid seems then to justify the regularization method, i.e., that parents should have a smaller norm than their children. However, this insight alone seems not particularly novel, as the same insight can be derived for standard hyperbolic method and has, for instance, been discussed in Nickel & Kiela (2017, 2018), Ganea et al (2018), De Sa (2018).”
>
> Although the fact that the representation of the common ancestor should have lower Euclidean norm is mentioned in these papers, it is never proven that it has lower Euclidean norm. The closest example that mentions a minimizer of an expectation of (squared) hyperbolic distances is De Sa [F]. However, they do not exploit a closed-form of the centroid and have to use a gradient-based method to minimize an optimization problem based on it (see [F], Section 4.2).
> We show that the Euclidean norm of the centroid of a set of point can be controlled with the curvature of the hyperbolic space. We experimentally show its impact in Table 2. Retrieval performance (Mean Rank and MAP) in Table 1 shows how close to its descendents a common ancestor is. The Poincaré metric is defined for a fixed curvature of -1 and cannot have smaller curvature given its formulation exploiting arcosh.
>
> Fig. 1 of our submission shows an example where the centroid of the Poincaré metric does not have a smaller Euclidean distance than the set of points.
> By manipulating the curvature of the space, the centroid of the Lorentzian norm can produce centroids with smaller Euclidean norm (as we demonstrate that they depend on each other).
> We can also plot the centroid of the squared Poincaré distance, which shows that the corresponding centroid does not have a smaller Euclidean norm either.
>
> - “p.3: Projection onto the Poincaré ball/manifold is only necessary when the exponential map isn't used. Nickel & Kiela (2018), Ganea et al (2018) therefore don't have this problem.”
>
> That is exactly what we explain in p.3, although Ganea et al. [E] also reproject their embeddings onto the Poincaré ball at each iteration (as explained in the “Numerical errors” paragraph of Section 4 of [E]).
> Nickel & Kiela (2018) propose to work in the hyperboloid space to avoid this reprojection as we explain in p.3.
>
> - “p.7: Since MR and MAP are ranking measures, and the ranking of distances between H^d and the L^2 distance should be identical, it is not clear why the experiments show significant differences for these methods when \beta=1”
>
> Although the order of distances is the same between H^d and the L^2 (since they only differ by an arcosh activation function), these two distance functions are not equivalent.
> The arcosh has a logarithmic form and then tends to penalize differences between small distances more than differences between large distances.
> This generates a difference during training that is for instance similar to the difference obtained by training a linear loss vs a quadratic loss. The quadratic loss tends to penalize outliers more than a linear loss.
> The fact that these distance functions are not equivalent explains the difference of the results.
>
> - “p.7: Embeddings in the Poincaré ball and the Hyperboloid are both compatible with the regularization method in eq.14. It would be interesting to also see results for these methods with regularization.”
>
> We agree but the point of that regularizer was to show that the global structure of the tree could be easily recovered by using such constraints without having a significant impact on the retrieval performances (i.e. Mean Rank and Mean Average Precision).
> We have added for instance in the updated version a study of the impact of such regularization on classification performance in Table 3. Removing that regularization consistently leads to (slightly) better classification scores.

---

### Official Review · AnonReviewer1 · 2018-11-06
**Review for Lorentzian Distance Learning**

**Rating:** 6
**Confidence:** 4

**Review:**

Summary

Learning embeddings of graphs in hyperbolic space have become popular and yielded promising results. A core reason for that is learning hierarchical representations of the graphs is easier in hyperbolic space due to the curvature and the geometrical properties of the hyperbolic space. Similar to [1, 2], this paper uses Lorentzian model of the hyperbolic space in order to learn embeddings of the graph. The main difference of the proposed approach in this paper is that  they come up with a closed-form solution such that each node representation close to the centroid of their descendants. A curious property of the equation for the centroids proposed to learn the embeddings of each node also encodes information related to the specificity in the Euclidean norm of the centroid. Also this paper introduces two additional hyperparameters. Beta hyperparameter is selected to control the curvature of the space. Depending on the task the optimal curvature can be tuned to be a different value. This also ties closely with the de-Sitter spaces. Authors provide results on different graph embedding benchmark tasks. The paper claims that, an advantage of the proposed approach is that the embedding of the model can be tuned with regular SGD without needing to use Riemannian optimization techniques.

Questions

Have you tried learning beta instead of selecting as a hyperparameter?
The paper claims that Riemannian optimization is not necessary for this model, but have you tried optimizing the model with the Riemannian optimization methods?
Equation 11, bears lots of similarity to the Einstein gyro-midpoint method proposed by Abraham Ungar which is also used by [2]. Have you investigated the relationship between the two formulations?
On Eurovoc dataset the results of the proposed method is worse than the d_P in H^d. Do you have a justification of why that happens?


Pros
The paper delivers some interesting theoretical findings about the embeddings learned in hyperbolic space, e.g. a closed for equation in the
The paper is written well. The goal and motivations are clear.


Cons
Experiments are only limited to small scale-traditional graph datasets. It would be more interesting to see how those embeddings would perform on large-scale datasets such as to learn knowledge-base embeddings or for recommendation systems.

Although the idea is interesting. Learning graph embeddings have already been explored in [1]. The main contribution of this paper is thus mainly focuses on the close-form equation for the centroid and the curvature hyperparameter. These changes provide significant improvements on the results but still the novelty of the approach is in that sense limited compared to [1].


Minor comment:

It is really difficult to understand what is in Figure 2 and 3. Can you reduce the number of data points and just emphasize a few nodes in the graph that shows a clear hierarchy.

[1] Nickel, Maximilian, and Douwe Kiela. "Learning Continuous Hierarchies in the Lorentz Model of Hyperbolic Geometry." arXiv preprint arXiv:1806.03417 (2018).
[2] Gulcehre, Caglar, Misha Denil, Mateusz Malinowski, Ali Razavi, Razvan Pascanu, Karl Moritz Hermann, Peter Battaglia et al. "Hyperbolic Attention Networks." arXiv preprint arXiv:1805.09786 (2018).

---

> ### Author Response · Authors · 2018-11-16
> **Response to AnonReviewer1**
>
> -“Eq 11, bears lots of similarity to the Einstein gyro-midpoint”
>
> As mentioned in our introduction, there exist various hyperbolic geometries with corresponding distances: the Poincaré distance, the Lorentzian distance and the gyrodistance [A,B]. We have investigated their relationships.
> To the best of our knowledge, the centroid of the Poincaré distance metric has no closed form solution. We provide in our submission a closed-form solution for the squared Lorentzian distance for any negative curvature of the space.
> The main motivation to introduce the centroid is our last theorem which shows that comparing distances wrt the squared Lorentzian distance with a set of points is equivalent to comparing distances with their centroid. By studying the properties of the centroid, we then have an idea of the behavior of distances with the corresponding set of points.
>
> Gulcehre et al. [C] optimize the Poincaré distance but exploit as representative the Einstein gyrocentroid, also known as the Einstein midpoint when the set has only 2 points. Unlike the 2 previous centroids, that point is in general not an optimizer of an expectation over gyrodistances to a set of points. Nonetheless, when the set contains only 2 points, it is the minimizer of the Einstein addition of the gyrodistances between it and the two points of the set by using the gyrotriangle inequality. However, the Einstein addition is not commutative, the expectation properties then do not generalize for a set of more than 2 points although the gyrocentroid does preserve left gyrotranslation.
>
> Conceptually, the main difference is that the Lorentzian centroid can be seen as a minimizer of some expectation over distances, the Lorentzian centroid is then ideal to represent the common ancestor of a set of tree nodes. On the other hand, the gyrocentroid can be seen as a point which preserves left gyrotranslation (see Remark 5.7 of [A]).  The motivation of exploiting the point that preserves left gyrotranslation to be compared wrt the Poincaré metric (which corresponds to another distance) is not clear to us, at least for our task.
>
> Each of the distances have different centroids which are illustrated in the updated Fig 1.
>
> - "Experiments are limited to small scale datasets."
>
> We report quantitative results on the same datasets as [D] and [E].
> More exactly, [E] consider two tasks but they mention for the first task: "for the sentence entailment classification task, we do not see a clear advantage of hyperbolic MLR compared to its Euclidean variant." We then compared our method in the task where they see an improvement when using hyperbolic representations. Our approach outperforms theirs.
>
> - "The paper claims that Riemannian optimization is not necessary for this model"
>
> We do not sell the fact that we do not use Riemannian optimization as a contribution. Other approaches such as [D] can also do that (although their function is not differentiable on the whole domain and gradients tend to infinity for pairs of points with infinitesimal distance). We only say that, given the formulation of our distance in Eq. (9), using standard SGD is sufficient to outperform current hyperbolic approaches.
> We have not tried a Riemannian optimizer since the performance of a standard SGD already works well. We plan to do that in the future.
>
> - "The novelty of the approach is limited compared to [1].”
>
> The main contribution of the paper is not only a closed-form solution for the centroid. We exploit our last theorem that explains that distances with a set of points can be reduced to the study of the distance wrt the centroid. We then analyze some of its properties and explain why they are appropriate to represent trees.
>
> - "Have you tried learning beta?"
>
> Following the review, we have learned beta by learning a variable constrained to be positive by using a softplus activation function.
> Here are results on some datasets, they are comparable with those reported when beta is 0.01:
> ACM: MR: 1.03 - MAP: 98.4 - SROC: 53.4
> EuroVOC: MR: 1.06 - MAP: 96.5 - SROC: 33.8
> Wordnet verbs: MR: 1.10 - MAP: 94.7- SROC: 26.1
> The learned beta has values in the interval [10^(-6),10^(-4)]
>
> - “On Eurovoc the results are worse than d_P in H^d”
>
> Only the SROC score is worse, we obtain better or comparable retrieval scores (i.e. MR and MAP) on this dataset. The SROC score can be improved by increasing the regularization parameter lambda, but we only reported scores for one value of lambda.
>
> - “Can you reduce the number of data points in Fig 2”
>
> As requested, we have added Fig. 5 that only plots the names of the kernel method researchers mentioned in Table 4. Most of them are represented close to the same radius, which validates our study of the Lorentzian distance for small values of beta.
> We have also added Fig. 3 to illustrate the Lorentzian distance to 1 point as a function of beta.

---

### Author Response · Authors · 2018-11-16
**We have updated the paper.**

We thank the reviewers, we clarify some points in the individual responses.

We have updated the paper with some requested additions.
The references that we use in our rebuttals are:

[A] Ungar, Analytic Hyperbolic Geometry in N dimensions
[B] Ungar, Barycentric Calculus in Euclidean and Hyperbolic Geometry, 2010
[C] Gulcehre et al., Hyperbolic attention networks. Arxiv 2018
[D] Nickel and Kiela, Learning continuous hierarchies in the lorentz model of hyperbolic geometry, ICML 2018
[E] Ganea et al., Hyperbolic neural networks. NIPS 2018
[F] De Sa et al., Representation tradeoffs for hyperbolic embeddings, 2018

---

### Meta-Review · Area_Chair1 · 2018-12-14
**A well-written paper that is a bit lacking in novelty**

**Confidence:** 3
**Recommendation:** Reject

**Metareview:**

Dear authors,

The reviewers all appreciated the treatment of the topic and the quality of the writing. It is rare for all reviewers to agree on this, congratulations.

However, all reviewers also felt that the paper could have gone further in its analysis. In particular, they noticed that quite a few points were either mentioned in recent papers or lacked an experimental validation.

Given the reviews, I strongly encourage the authors to expand on their findings and submit the improved work to a future conference.